# Simulation and Experimental Study on Stress Relaxation Response of Polycrystalline γ-TiAl Alloy under Nanoindentation Based on Molecular Dynamics

**DOI:** 10.3390/mi15081020

**Published:** 2024-08-09

**Authors:** Junye Li, Chunyu Wang, Jianhe Liu, Xiwei Dong, Jinghe Zhao, Ying Chen

**Affiliations:** 1Ministry of Education Key Laboratory for Cross-Scale Micro and Nano Manufacturing, Changchun University of Science and Technology, Changchun 130022, China; ljy@cust.edu.cn (J.L.); wcymall@163.com (C.W.); ljhmach@163.com (J.L.); cjmsys@sina.cn (X.D.); 2School of Mechanical Engineering, Changchun Guanghua University, Changchun 130033, China; 3Jilin Province Product Quality Supervision and Inspection Institute, Jilin 130023, China; w438031480@163.com

**Keywords:** nano-indentation, polycrystalline γ-TiAl alloy, stress relaxation, molecular dynamics simulation, experimental study

## Abstract

This study employed nano-indentation technology, molecular dynamics simulation, and experimental investigation to examine the stress relaxation behaviour of a polycrystalline γ-TiAl alloy. The simulation enabled the generation of a load-time curve, the visualisation of internal defect evolution, and the mapping of stress distribution across each grain during the stress relaxation stage. The findings indicate that the load remains stable following an initial decline, thereby elucidating the underlying mechanism of load change during stress relaxation. Furthermore, a nano-indentation test was conducted on the alloy, providing insight into the load variation and stress relaxation behaviour under different loading conditions. By comparing the simulation and experimental results, this study aims to guide the theoretical research and practical application of γ-TiAl alloys.

## 1. Introduction

The demand for high-temperature-resistant structural materials in contemporary aerospace, shipbuilding, vehicle manufacturing and other industrial fields has led to a growing interest in TiAl-based alloy materials. In the 1950s, preliminary development and study of Al compounds of Ti were initiated. By the 1970s, the advent of the global energy crisis led to a gradual shift in focus towards the Al compounds of Ti. In recent decades, TiAl alloys have been the subject of considerable research interest due to their superior properties compared to traditional superalloys. These include a high melting point, low density, high specific strength, corrosion resistance, excellent oxidation resistance and good structural stability [1,2,3,4,5]. As the field of application for γ-TiAl alloys continues to expand, the working environment in which the material is used becomes increasingly complex. Consequently, the service performance and reliability of γ-TiAl alloys are subject to higher standards [6,7]. Researchers now consider not only the overall strength, stiffness, and stability of the macroscopic workpiece but also the surface mechanical properties and microstructure changes in the material exposed to the working environment.

In recent years, the in-depth study of the surface properties of materials has led to the continuous exploration of the microscopic world. This has resulted in the traditional material mechanics test method being unable to meet the needs of today’s material testing. Nanoindentation technology [8] represents a novel approach to characterising the properties of materials. This method is employed by researchers to investigate the mechanical properties of materials at the micron and nanometre scales. Nevertheless, it remains a significant challenge to study the instantaneous atomic displacement and structural changes in materials during nanoindentation experiments. The molecular dynamics simulation method allows for the simulation of the interaction between the indenter and the material at the atomic scale, the analysis of the evolution of the micro-defect structure, and the acquisition of the mechanical properties and deformation mechanism. Consequently, it offers a robust theoretical foundation for practical applications. Zhang, Y. conducted an explicit implementation of the nonlocal operator method, specifically the nonlocal dynamic equation of elastic solids [9]. Mondal, D. and other scholars conducted a comparative study of the VMD denoising method based on Monte Carlo simulation. This study compared the newly introduced method based on autocorrelation with the method based on PDF distance [10]. In their study of self-supervised learning for tool wear monitoring, von Hahn et al. employed a disentangled variational autoencoder [11]. Zhao et al. used molecular dynamics to simulate the deformation anisotropy of materials during polishing [12]. The research offers robust theoretical justification for practical applications. The material will undergo specific deformation as a result of the application of an external force, and a certain force is required to maintain the deformation. The stress relaxation process is maintained for a period of time under fixed deformation and thus represents an area of interest for many scholars engaged in the study of materials. Chen et al. [13,14,15] employed the molecular dynamics method to investigate the stress relaxation behaviour of monocrystalline silicon coated with amorphous SiO_2_ films during nanoindentation. It is observed that alterations in the maximum indentation depth, loading speed, and indenter size result in an increase in stress relaxation. The relaxation is found to be the greatest when the maximum indentation depth, loading speed, and indenter size are at their highest. Varma et al. [16] conducted a stress relaxation test on the SS316L material during the tensile process and observed that the stress relaxation process enhances the ductility of the SS316L material. In a study conducted by Nguyen et al. [17], stress relaxation tests were performed on high-strength steel wires. The research aimed to investigate the structural evolution, stress relaxation behaviour, and the effect of stress relaxation on mechanical properties. A substantial body of research has been conducted on the mechanisms of γ-TiAl alloys during plastic deformation [18,19,20,21,22,23]. However, there is a paucity of literature on the behaviour of the alloy under stress relaxation.

In conclusion, the nanoindentation response of a polycrystalline γ-TiAl alloy was investigated through molecular dynamics simulation. Furthermore, the stress relaxation behaviour of the same alloy was also subjected to analysis. By employing molecular dynamics simulation in conjunction with the load-time curve during the stress relaxation phase, the progression of defects and the distribution of stress within each grain can be elucidated. This also elucidates the micro-deformation mechanism of the polycrystalline γ-TiAl alloy under stress relaxation. The micro-nano test system was employed to elucidate the variation in the stress relaxation behaviour of the γ-TiAl alloy under different loads. A comparison between the reliability of the experimental and simulation results has been established, providing a reference point for further exploration of the mechanical properties and plastic deformation mechanism of the γ-TiAl alloy through molecular dynamics simulation.

## 2. Simulation Study of the Nano-Indentation Response in the Stress Relaxation Process

### 2.1. Model Building

This study employed molecular dynamics simulation to investigate the stress relaxation behaviour of a polycrystalline γ-TiAl alloy. The Voronoi algorithm was employed to generate polycrystalline workpieces with dimensions of 220 Å × 220 Å × 220 Å and comprising six grains. The workpiece model is depicted in Figure 1, where G1–G5 represents the grain number at different locations. The Berkovich diamond indenter is employed for the indentation operation, with the indenter tip positioned at the red point indicated in Figure 1a. Figure 1b illustrates that the overindentation point is aligned with the YZ plane. It can be observed that both G1 and G2 exhibit indentation on both sides of the point of contact, while G5 is situated below the grain boundary between G1 and G2. The pressure head should be positioned 10 Å away from the upper surface of the workpiece, as the hardness of the diamond indenter is significantly higher than that of the alloy. In the simulation, the indenter is regarded as a rigid body, and the interaction force of its internal atoms is ignored [20,24]. The workpiece is divided into three distinct regions: the boundary layer, the constant temperature layer, and the Newtonian layer. The boundary layer is employed to fix the workpiece, the constant temperature layer is used to transfer heat and ensure the heat exchange of the system, and the Newtonian layer is the indentation region. In this paper, a model is developed that generates nodes at random locations inside the simulation box and randomly orientated grains for the region where each node is located. This results in a polycrystalline model. This approach has no impact on the analysis of the deformation process of the model for γ-Ti-Al. The model comprises a total of 478,154 particles, with the green particles representing the fcc lattice structure and accounting for 83.4% of the total. The white particles represent the other lattice structures and account for 16.6% of the total. The polycrystalline γ-TiAl alloy nano-indentation molecular dynamics simulation model established in this paper is depicted in Figure 2.

### 2.2. Molecular Dynamics Principle and Potential Function Selection of Nanoindentation

#### 2.2.1. Selection of Potential Function

The potential function can characterize the interaction between the particles of each part of the material, which is the basis of molecular dynamics simulation. In the process of molecular dynamics simulation, the relationship between the various parts needs to be described by appropriate potential functions, which is related to the calculation time, calculation efficiency and accuracy of the simulation results. The research in this paper involves a Berkovich indenter with a diamond structure and a γ-TiAl alloy. The specific potential functions are as follows:(1)EAM potential function

The embedded atom method (EAM) potential function [25] is based on density functional theory, which can accurately and reasonably describe the metal and alloy systems and is widely used. In this paper, the EAM potential function is used to describe the interaction between Ti-Al alloys. The specific expression of the total atomic energy *E* is
(1)$$E=\sum_{i}F_{i}(\rho_{i})+\frac{1}{2}\sum_{i\neq j}\phi_{ij}(r_{ij})$$

In the formula, *F_i_* denotes the embedding energy function of atom *i*, which is the sum of the density of the electron cloud generated by the extranuclear electrons of atoms other than the *i* th atom at the *i* th atom. Is the pair potential interaction function between atoms *i* and *j*; represents the distance between atoms *i* and *j*.
(2)Morse potential function

As shown in Table 1, the Morse potential function [26] is a typical pair potential based on the two-atom theory. It is analytically simple and has a wide range of applications. It can be effectively applied to the deformation of metals. Therefore, in this paper, the Morse potential function is used to describe the Ti-C and Al-C interaction between the indenter and the workpiece. The energy *E* of the two atoms *i* and *j* with a distance is given in the following formula:(2)$$E=D_{0}\left[e^{-2\alpha (r_{ij}-r_{0})}-2e^{-\alpha (r_{ij}-r_{0})}\right]$$

In the formula, *D*_0_ is the binding energy coefficient, and α is the gradient coefficient of the potential energy curve, which is the equilibrium distance between two atoms. The Morse potential function parameters between Ti, Al, and C atoms [27] are shown in Table 2.


**Table 1 micromachines-15-01020-t001:** Morse potential function parameters.

Unit	Table	Al-C
*D*_0_ (eV)	0.9820	0.2800
*α* (Å^−1^)	2.2830	2.7800
*r*_0_ (Å)	1.8920	2.2000

**Table 2 micromachines-15-01020-t002:** Simulation conditions and parameters of nano-indentation.

Simulation Condition	Simulation Parameters
Workpiece material	Polycrystalline γ-TiAl alloy
Number of grains of workpiece	6
Atomic number of workpieces	631,504
Shape and material of indenter	Berkovich Diamond indenter
Head atomic number	18,596
Time step	1 fs
Indentation speed	50 m/s
Stress relaxation time	50 ps
Lifting speed	50 m/s
Simulated temperature	293 K
Potential function	EAM, Morse

#### 2.2.2. Equilibrium Ensemble of Molecular Dynamics Simulation

In the context of molecular dynamics simulation, the control of particles must be conducted within a defined ensemble. The selection of an appropriate ensemble for simulation is dependent upon the specific system under investigation and the requisite information for calculation. The selection of an appropriate ensemble is crucial for ensuring the reliability of the simulation and for reducing the time required for its completion. The simulation process described in this study does not entail a change in the number of particles, and the simulation system is a closed system in which the number of particles remains constant. The most commonly employed ensembles are the micro-canonical ensemble, the canonical ensemble, and the constant-temperature, constant-pressure ensemble.
(1)Microcanonical ensemble

The microcanonical ensemble is also referred to as the NVE ensemble, which maintains the number of particles (N), volume (V), and energy (E) at a constant value. The simulation can be conducted in such a way that the number and volume of particles remain constant throughout. Furthermore, there is no change in the volume and number of particles during the course of the simulation. Concurrently, the system is isolated from the external environment, thereby maintaining a constant total energy state.
(2)Canonical ensemble

The canonical ensemble represents a set of many systems with the same temperature. This maintains the number of particles (N), volume (V), and temperature (T) within the system, which is also known as the NVT ensemble. From a practical standpoint, there is a greater focus on changes in temperature than on energy. Therefore, the temperature, volume, and particle number are controlled in the NVT ensemble. In canonical ensembles, a variety of techniques are employed to maintain system temperature stability. These include the velocity scaling method, the thermal bath coupling method, the Anderson thermal bath method, and the Nose–Hoover thermal bath method.
(3)Constant-temperature and constant-pressure ensemble

In the constant-temperature and constant-pressure ensemble, the number of particles (N), pressure (P), and temperature (T) are maintained at constant values, which is also referred to as the NPT ensemble. The ensemble permits alterations in the volume of the system and enables energy exchange with the external environment, while maintaining a constant overall temperature and pressure. In this ensemble, there are two ways to alter the volume: one is to modify the size while maintaining the shape, and the other is to change both the size and shape simultaneously.

The nanoindentation simulation process described in this paper is divided into three distinct stages: a relaxation stage, a loading pressure stage, and an uplift stage. The relaxation process serves to equilibrate the initial structure model, thereby ensuring that the system reaches a equilibrate and uniform state. This prevents the unreasonable initial structure from leading to the failure of the simulation results. The relaxation stage, as simulated in this paper, employs a Nose–Hoover hot bath to achieve equilibrium. Following a period of temperature-controlled adjustment, the initial structure model reaches a state of equilibrium. During the loading and upward phases of the simulation, the temperature and energy of the system fluctuate in accordance with the simulation process, while the volume remains largely constant. Consequently, the system will be simulated under the NVE ensemble.

In order to accurately express the macroscopic properties of the material in the X and Y directions, periodic boundary conditions are employed. By contrast, in the Z direction of the indentation, the free boundary condition is employed to permit the atoms on the surface of the workpiece to migrate along the indentation direction. The diamond indenter is pressed along the Z direction at a speed of 50 m/s. Once the indentation depth reaches 1.5 nm, the stress relaxation simulation is initiated by maintaining the indenter displacement at a constant value. The holding time is identical to the duration of the indenter pressing process, after which the indenter returns to its original position at a speed of 50 m/s. Table 2 lists the parameters used to simulation conditions and parameters of nano-indentation.

### 2.3. Response Analysis of Pressure Relaxation Process

In order to study the stress relaxation behaviour and deformation mechanism of polycrystalline γ-TiAl alloy under constant strain, and to explore the load change and the internal microstructure change in the workpiece during the displacement holding stage, a nano-indentation response analysis of polycrystalline γ-TiAl alloy during the loading–holding–unloading process was carried out. The load and time variation in the polycrystalline workpiece during stress relaxation was obtained, as shown in Figure 3.

As shown in Figure 3, the load-time curve of polycrystalline γ-TiAl with a depth of 1.5 nm is obtained by keeping the indenter displacement unchanged between the loading process and the unloading process. Upon reaching 50 ps, the depth reaches 1.5 nm, at which point the simulation enters the displacement holding phase. At 100 ps, the simulation enters the unloading stage. As illustrated in Figure 3, following the conclusion of the loading phase, the load exhibits a gradual decline and appears to reach a state of equilibrium. When the indenter acts on the polycrystalline workpiece, the atoms at the grain boundary are more active, and the frequency of the atomic impact indenter is higher in the early stage of stress relaxation. Consequently, the load decreases slowly when the polycrystalline workpiece enters the stress relaxation stage. Concurrently, the workpiece exhibits a large vibration, and then the load gradually remains stable. The average load between 60 and 100 ps was found to be stationary, with a value of 446.34 nN. The maximum load was observed at 50 ps, with a value of 639.03 nN. The load decreased by 192.69 nN during the stress relaxation stage of the polycrystalline workpiece. Upon reaching 100 ps and entering the unloading stage, the load value exhibits a gradual decrease, reaching zero at 116 ps. In order to ascertain the cause of the load drop of the polycrystalline workpiece during the displacement holding stage and the changes in grains, grain boundaries, dislocations, and stacking faults in the workpiece, a defect structure distribution and evolution in the workpiece was studied by DXA analysis. The results of this study are presented in Figure 4.

Upon reaching 50 ps, the simulation process transitions to the stress relaxation stage. At this juncture, the displacement of the indenter remains unaltered, and the indenter is no longer capable of destroying the workpiece. At this juncture, the grain designated as G2 is most significantly impacted. At 53 ps, as illustrated in Figure 4a, the internal defect structure of G2 is annihilated and releases energy, resulting in the formation of a small dislocation line comprising white grains at the grain boundary, which surround the stacking fault. Within G1, Lattice a, b, and c layer errors change law, as shown in Figure 4, and layer error b loses the supply of energy—no longer on layer error a and layer error c—to expand the influence of layer error a and layer error c; layer error a and layer error c are expanded in the direction of lower energy for slip, to achieve a stable state. Figure 4b,c demonstrate that the internal defect structures of G1 and G2 remain stable throughout the relaxation stage. The load fluctuates considerably during this period. This is attributable to the elevated atomic activity of the polycrystalline grain boundary, which in turn affects the constant displacement of the indenter. As illustrated in Figure 4d, with the indenter’s ascent, the exogenous accumulation fault c in G1 is progressively incorporated into the grain boundary between G1 and G4. Concurrently, the accumulation fault a shifts in a parallel trajectory to the right. As illustrated in Figure 4e, at 130 ps, the indenter has fully absorbed the accumulation fault c, resulting in a significant reduction in the area of the accumulation fault a. As illustrated in Figure 4f, the stacked fault a persists within G1, with the attachment of shorter dislocation curves to the grain boundaries. Upon the indenter’s departure from the interior of G2, the grain boundary atoms between G1 and G2 form a minor stacking fault structure, situated within the confines of surrounding grain boundaries.

During the loading and unloading process of G3 grains, it was observed that small stacking faults would not form stable lattice nucleation following the stress relaxation stage. This was due to the fact that the energy existing in G3 grains was released at this stage, preventing the formation of stable lattice nucleation. It can thus be concluded that the energy available is insufficient to support the formation of stacking fault structures in G3 grains. With regard to G4 grains, it can be observed that the internal microstructure undergoes a further transformation during the stress relaxation stage. Figure 5 illustrates the evolution of the internal microstructure of G4 grains. By contrast, for other grains, the limited indentation depth prevents the indenter from providing sufficient energy to penetrate the grain boundary, resulting in the formation of defects in neighbouring grains.

As illustrated in Figure 5a, the G2 grains are sectioned along the A-A interface, and the slip curves in three directions (1, 2 and 3) are generated in the A-A section, resulting in the formation of stacked faults. The internal defects in the G2 grains exhibit a notable reduction in structural complexity, accompanied by a reduction in volume. Conversely, the G4 grains display an increase in volume. The released energy is transferred to the G4 grains via the grain boundaries, thereby promoting the generation of stacking faults in the G4 grains. As illustrated in Figure 5a, the initial stacking fault 1 structure within the G4 grain undergoes expansion at 53 ps, resulting in the emergence of two new stacking fault 2 and 3 structures within the stacking fault. One side of the stacking fault 2 is driven by a Shockley partial dislocation to expand, and the stacking fault 3 is surrounded by grain boundaries without dislocation, thereby becoming a stable structure. The movement of dislocations and the development of stacking faults significantly facilitate the transfer and release of stress through grain boundaries, thereby reducing the load applied by the indenter. As illustrated in Figure 5b, the internal defect structure of G4 becomes increasingly stable over time. At a time of 100 ps, as illustrated in Figure 5c, the compressive rod dislocation forms a dislocation junction with two minor Shockley dislocations. This limits the further evolution of the internal defect structure of the workpiece during the stress relaxation stage. As illustrated in Figure 5d, the indenter commences the unloading stage, accompanied by an increase in activity among the grain boundary atoms. The energy begins to be released, and the area of the stacked fault 1 gradually diminishes. At this juncture, the bar dislocation undergoes a gradual reduction in length, while the Shockley dislocation exhibits a corresponding increase. At 110 ps, as illustrated in Figure 5e, the compressive rod dislocation is annihilated, and the right side of stacking faults 1 and 2 is surrounded by Shockley dislocations. Ultimately, as illustrated in Figure 5f, the stacking fault 2 expands and is incorporated into the grain boundary, thereby forming a stable structure. As a consequence of the complete loss of pressure head in the Ti-Al cell, the energy at the grain boundary is fully released, resulting in a slight expansion of the accumulation fault 1 and its retention within the cell, which in turn gives rise to structural deformation.

In summary, at the beginning of the stress relaxation stage, the defect structure inside the grain continuously impacts each grain boundary, resulting in energy transfer and release. On the one hand, the atoms at the grain boundary have large movable space and strong load capacity, which reduces the impact of the workpiece atoms on the indenter and reduces the indenter load. On the other hand, the atomic activity at each grain boundary is strong, which constantly has a certain degree of impact on the indenter, so the load fluctuation in the stress relaxation stage is relatively large. In order to study the influence of stress relaxation process on the internal stress distribution of the workpiece, the cross section of the same position in Figure 5a is obtained, and the stress distribution at different times is shown in Figure 6.

The load-time curve and defect evolution in the stress relaxation stage indicate that when the loading process is completed and the stress relaxation is completed, the load is reduced, and the defect structure inside the workpiece is partially annihilated. Among the factors contributing to this phenomenon is the stress change in the workpiece when the indenter displacement is maintained. At 53 ps, as illustrated in Figure 6a, the high-stress zone diminishes towards the area proximate to the indenter, and the internal stress of the workpiece is significantly reduced, resulting in the annihilation of the defect structure in grain G2. As time continues to elapse, as illustrated in Figure 6b, the stress within the workpiece is released, the stress concentration disappears, and the residual stress maintains the existence of a stable defect structure. At 100 ps, as illustrated in Figure 6c, the stress distribution within the workpiece is comparable to that observed at 100 ps in Figure 6b, indicating that during the period of steady load, the stress distribution within the workpiece remains stable, thus maintaining the internal defect structure of the workpiece. At 105 ps, as illustrated in Figure 6d, following stress relaxation, the internal stress of the workpiece dissipates more rapidly in the unloading stage. In particular, the stress near the grain G2 returns to its initial level, and the defect structure inside G2 is annihilated without stress support. Conversely, a small stress-supported stacking fault structure persists in G1. At 110 ps, as illustrated in Figure 6e, the stress within the workpiece gradually decreases to its initial level. In the G1 region, the absence of stress due to the restriction of the grain boundary prevents the evolution of the defect structure, which remains within the workpiece. As the pressure head continues to rise, as illustrated in Figure 6f, the stress within the workpiece remains relatively constant, indicating that the activity of atoms within the workpiece is limited and that the existing microstructure of the workpiece is maintained.

## 3. Experimental Study on Nano-Indentation Response in the Stress Relaxation Process

### 3.1. Test Preparation

The addition of the Al element represents a significant enhancement to titanium alloys, as it serves to reduce both the density and melting point of the alloy. This is advantageous for the production of lightweight alloys and enhances the room temperature and high temperature strength of the alloy. As the Al content is increased, the phase composition of the alloy gradually changes in order to obtain different types of Ti-Al intermetallic compounds. As illustrated in the Ti-Al binary phase diagram (Figure 7), Ti_3_Al, TiAl, and TiAl_3_ are the most prevalent. Of these, alloys of the A_3_B and AB types, namely Ti_3_Al and TiAl, have significant potential for application and development as high-temperature structural materials. Ti_3_Al and TiAl-based alloys have developed rapidly in response to the robust demand for high-temperature structural materials that are stronger, more rigid, more heat-resistant, and lighter in weight, which are required for use in the aerospace, marine, automotive, and other fields.

According to the phase diagram of Ti-Al alloy in Figure 7 [28,29], the polycrystalline γ-TiAl alloy was customized by vacuum magnetic levitation melting method according to the atomic ratio of Ti to Al at 1:1. Al is a substitutional α-phase stabilizer and is considered the main alloying element in α and α+β titanium alloys. The addition of Al can increase the β-transus temperature and strengthen the α phase by substitutional solid-solution strengthening. Also, Al has a large solubility in titanium. The Ti-Al binary phase diagram is shown in Figure 7. By increasing the Al content, different intermetallic compounds can be formed, including Ti_3_Al, TiAl, and TiAl_3_. The early studies of titanium alloys, with aluminium concentration above 6 wt.%, confirmed the ordered Ti; Al (α) phase under certain heat treatment conditions. The α phase shows the hexagonal DOig crystal structure, where Ti and Al atoms occupy particular locations in the hexagonal close-packed (HCP) structure [30].

In order to guarantee the precision of the subsequent micro-nano indentation test, an EDS energy spectrum analysis of the workpiece was conducted prior to the test, enabling the composition to be identified. Figure 8 below illustrates the energy spectrum diagram of the material composition, while Table 3 presents the content analysis of each component element of the material. In elemental content analysis, “Sigma” usually refers to Standard Deviation, which is a statistical quantity used to describe the degree of dispersion of a set of data, i.e., the deviation of the data values from the mean value. In the elemental content analysis report, Sigma (Standard Deviation) is one of the important indexes for evaluating data quality, analysis accuracy, and reliability. On the whole, the γ-TiAl alloy sample meets the testing requirements of this experiment.

The diameter of the workpiece employed in this study is 15 mm × 5 mm, and it is then embedded. In the grinding machine, 100, 200, 400, 600, 800, 1000, 1200, 1500, and 2000 mesh SiC sandpaper is employed for grinding. Subsequently, the polishing machine is utilised to apply polishing paste, thereby enhancing the surface quality of the workpiece and reducing the impact of surface roughness on the precision of micro-nano indentation test outcomes. Following the grinding and polishing stages, the workpiece is ultrasonically cleaned in alcohol to remove surface contaminants and then dried in air. As illustrated in Figure 9, the processed workpiece has been polished to a mirror-like finish and meets the requisite standards for a nanoindentation test.

As illustrated in Figure 10, the test instrument utilized in this study is the micro-nano mechanics testing system produced by the FemtoTools company in Buchs, Switzerland. The specific model is the FT-MTA03, which is capable of measuring forces within a range of ±200 nN with a mechanical resolution of 0.5 nN. The indenter employed is the FT-S200000 micro force sensor from Hamburg, Germany, FEM, which has a mechanical range of ±200,000 μN and a mechanical resolution of 500 nN. The indenter is a Berkovich pyramid, constructed from diamond. In a nano-indentation test, both the sample and the measurement system undergo deformation. This results in a combined deformation of the sample and the system. In order to ensure the accuracy of the measurement data, it is essential to calibrate and compensate for any potential measurement errors prior to the commencement of the test. Prior to the commencement of the nano-indentation test, the instrument is calibrated and validated using quartz standard parts with a smooth surface. This ensures that the instrument is capable of meeting the experimental requirements.

The test probe employed in the test process delineated in this chapter is illustrated in Figure 11. The model is the FT-S200000 micro-force sensor, which has a mechanical range of ±200,000 μN and a mechanical resolution of 500 nN. The probe indenter is of the Berkovich triangular pyramid shape and is composed of diamond.

### 3.2. Experimental Study on Nano-Indentation Response during Stress Relaxation

In order to study the loading condition of the γ-TiAl workpiece without deformation, this indentation test maintains the displacement after the end of loading, and then observes the stress condition. This provides the analysis basis for the stress condition of the workpiece in the actual situation. The nano-indentation test is conducted using a displacement control method.

In the experiment, the maximum indenter load was maintained at a constant level. Five groups of experiments were conducted, with loads of 10 mN, 15 mN, 20 mN, 25 mN, and 30 mN, and a stress relaxation nanoindentation experiment was performed. The loading speed was set at 50 m/s, and each indenter load variable within the stress relaxation group was subjected to testing. The position of the indenter was maintained for a period of 25 s during the stress relaxation phase. The displacement control mode was employed throughout the process to maintain a constant indenter loading rate during loading and unloading, in accordance with the methodology used in the MD simulation.

Figure 12 illustrates how the load increases gradually throughout the loading process, reaching its maximum value at the conclusion of the loading phase. Due to the hysteresis inherent in the control of the equipment, the load may exceed a preset maximum load. Upon entering the displacement holding stage, the load decreases rapidly, and then enters the stable stage. During this period, a small increase in the load is observed, which is attributed to the drift of the equipment. This results in the load dropping below the load value when the displacement is maintained. A comparison between the simulation results and the test data indicates that at the initial stage of stress relaxation, the energy provided by the indenter to the workpiece is no longer increased. Instead, the defect structure inside the workpiece is further produced and annihilated, resulting in the release of stress and a decrease in the load on the indenter. The average load of the stable stage is taken as the load value of the displacement holding phase, which is recorded together with the maximum load in Table 4. In the unloading stage, the load decreases rapidly and reaches zero. Figure 12 illustrates that there is a discrepancy between the load required by the workpiece to maintain a constant displacement and the preset load. This implies that a smaller load can achieve the same displacement as the preset load. This phenomenon aligns with the load-time curve observed in the simulated stress relaxation stage.

Table 4 illustrates the discrepancy between the maximum load and the load in the displacement holding phase under varying loading conditions. Figure 13 offers a more intuitive representation of the relationship between the aforementioned discrepancy and the preset load.

In the load holding stage, the pressure depth gradually increases with time. This is due to the load holding method used in the test, the load remains unchanged, and the pressure depth is deepened to keep the load within a certain range. By calculating the displacement difference before and after the load holding stage, the displacement difference ballast line diagram shown in Figure 13 is drawn. With the increase in ballast, the displacement difference increases significantly. This is because the larger the load, the more stress is released during the load holding stage, and these stresses require a larger indenter displacement to keep the load stable within the set maximum load range.

## 4. Conclusions

This paper presents a molecular dynamics simulation of the nano-indentation process of a polycrystalline γ-TiAl alloy at room temperature. The load-time curve, defect evolution, and stress distribution are analysed. Concurrently, a nano-indentation test was conducted on a polycrystalline γ-TiAl alloy workpiece, resulting in the acquisition of a load-time curve through the stress relaxation process at room temperature. By comparing the results of the simulated and tested load versus time curves, the following conclusions were drawn from the similarity of the trends:(1)The experimental study of the stress relaxation stage revealed that the load change trend aligns with the findings of the molecular dynamics simulation. Specifically, the load initially decreases and then stabilizes when the displacement of the indenter remains constant. As the loading load increases, the discrepancy between the maximum load value attained during loading and the load value maintained during the displacement holding stage gradually widens. Upon application of a significant load, the γ-TiAl workpiece attains a specific degree of deformation and subsequently maintains this deformation. The degree of stress relaxation in the workpiece is considerable, as evidenced by the load-time curve, which illustrates a pronounced decline in load. The load change trend in the two is consistent with time, thereby elucidating the response mechanism of the Ti-Al alloy in the stress relaxation stage.(2)In the stress relaxation stage, the discrepancy between the maximum load value attained during loading and the load value sustained in the displacement holding stage gradually increases with the rise in loading. The γ-TiAl workpiece exhibits a high degree of deformation even under large loads, accompanied by a significant relaxation of internal stresses. This is evidenced by the load-time curve, which shows a pronounced decline in the load value. The load-time curve of the stress relaxation process obtained by the simulation process is identical to that obtained by the simulation, thereby providing reliable support for molecular dynamics simulation to study the characteristics of an actual γ-TiAl alloy, resolve the issue of room temperature brittleness, and enhance the performance of TiAl-based alloys.

## Figures and Tables

**Figure 1 micromachines-15-01020-f001:**
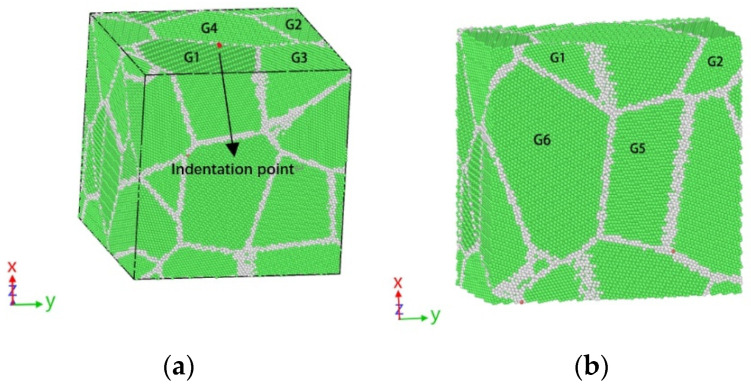
Workpiece model of the polycrystalline γ-TiAl alloy. (**a**) Crystal lattice structure diagram. (**b**) Sectional view of lattice structure.

**Figure 2 micromachines-15-01020-f002:**
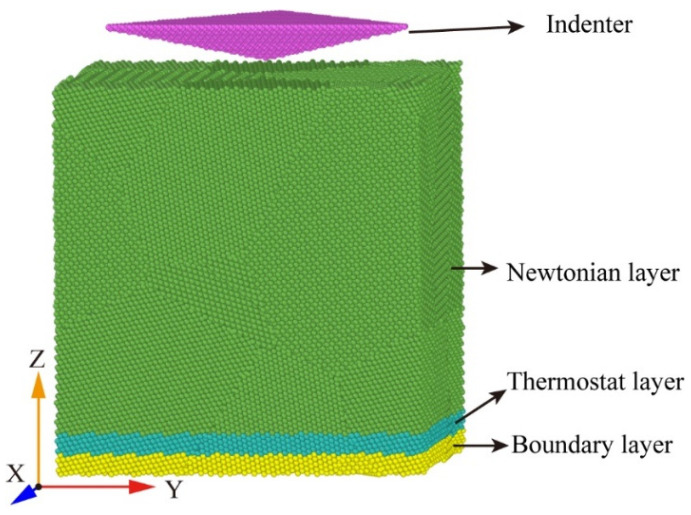
Molecular dynamics simulation model of nano-indentation for the polycrystalline γ-TiAl alloy.

**Figure 3 micromachines-15-01020-f003:**
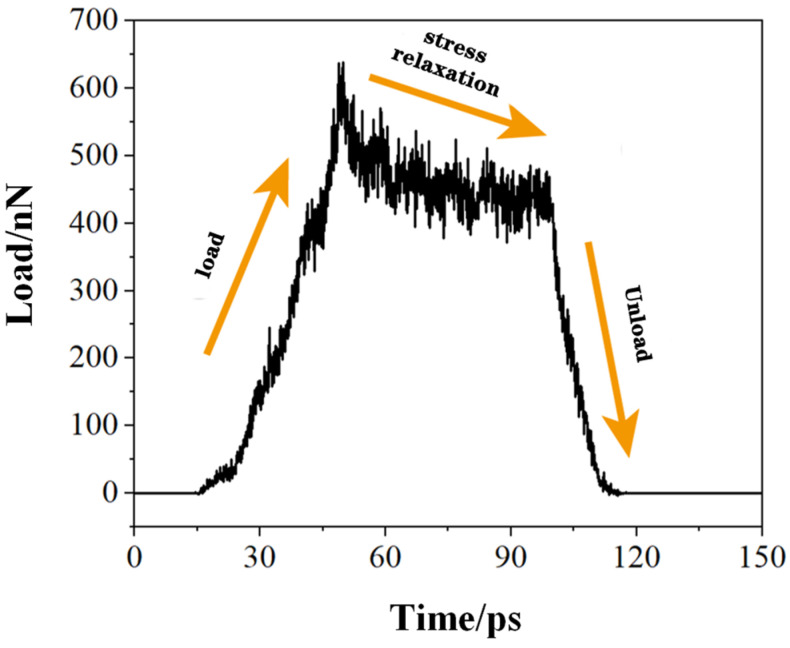
The load-time curve.

**Figure 4 micromachines-15-01020-f004:**
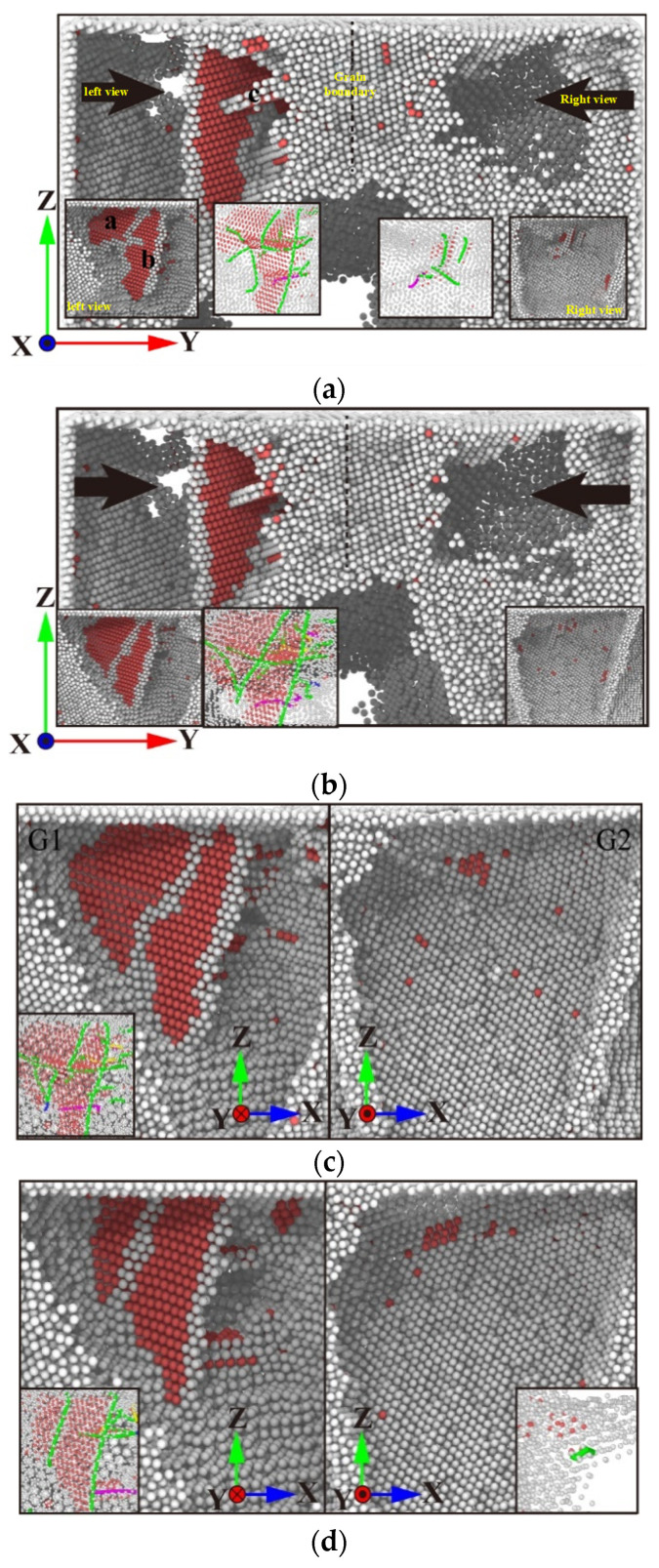
Distribution and evolution of defect structure in workpiece during stress relaxation. (**a**) Time 53 ps, (**b**) time 70 ps, (**c**) time 100 ps, (**d**) time 110 ps, (**e**) time 130 ps, and (**f**) time 150 ps.

**Figure 5 micromachines-15-01020-f005:**
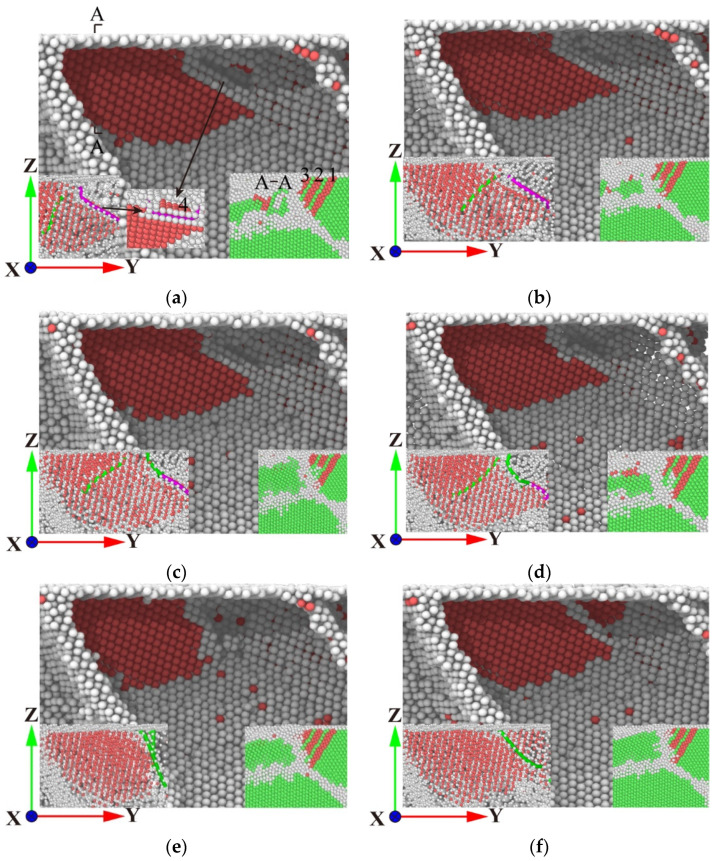
Microstructure evolution of G4 grains during stress relaxation. (**a**) Time 53 ps, (**b**) time 70 ps, (**c**) time 100 ps, (**d**) time 105 ps, (**e**) time 110 ps, and (**f**) time 150 ps.

**Figure 6 micromachines-15-01020-f006:**
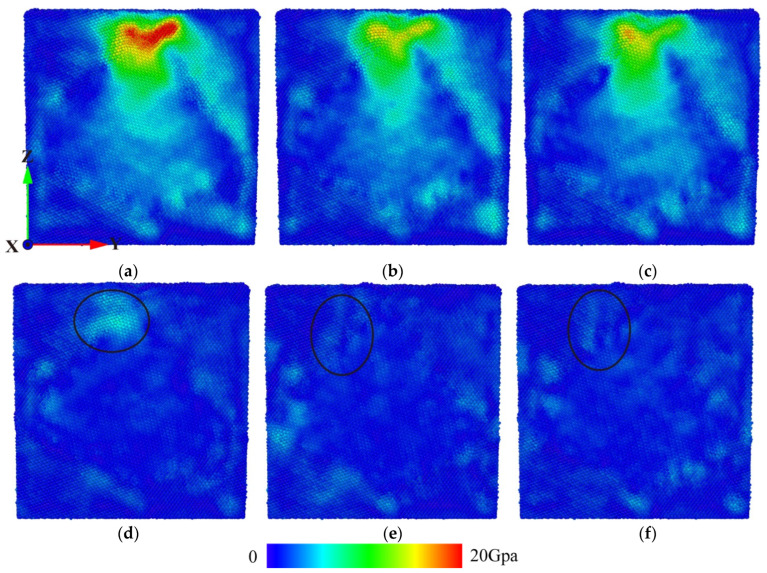
The equivalent stress distribution in the workpiece during stress relaxation. (**a**) Time 53 ps, (**b**) time 70 ps, (**c**) time 100 ps, (**d**) time 105 ps, (**e**) time 110 ps, and (**f**) time 150 ps.

**Figure 7 micromachines-15-01020-f007:**
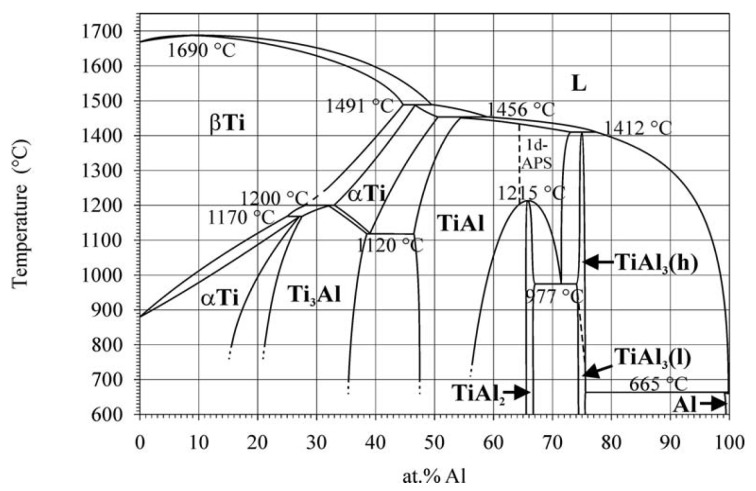
Ti-Al system according to the current fraction assessment.

**Figure 8 micromachines-15-01020-f008:**
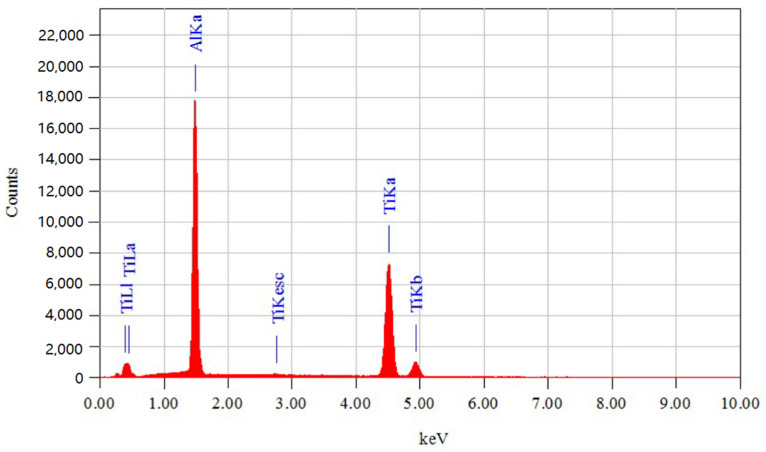
The EDS energy spectrum analysis.

**Figure 9 micromachines-15-01020-f009:**
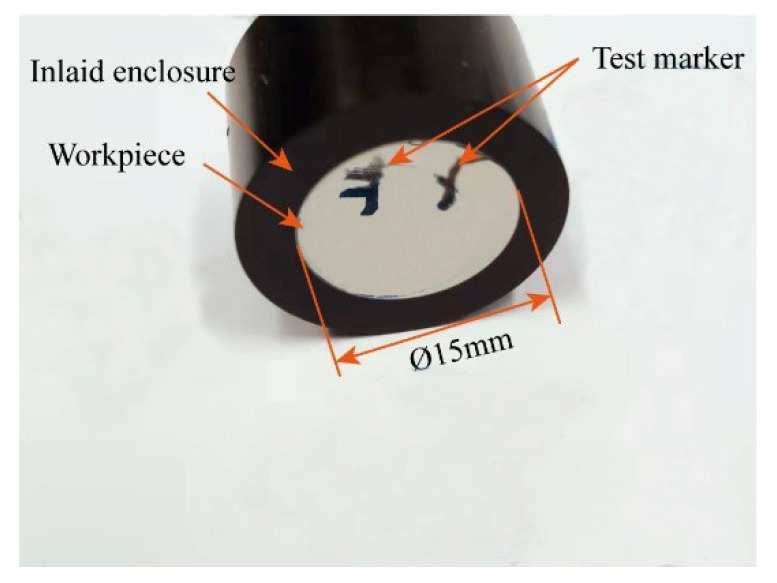
Test workpiece.

**Figure 10 micromachines-15-01020-f010:**
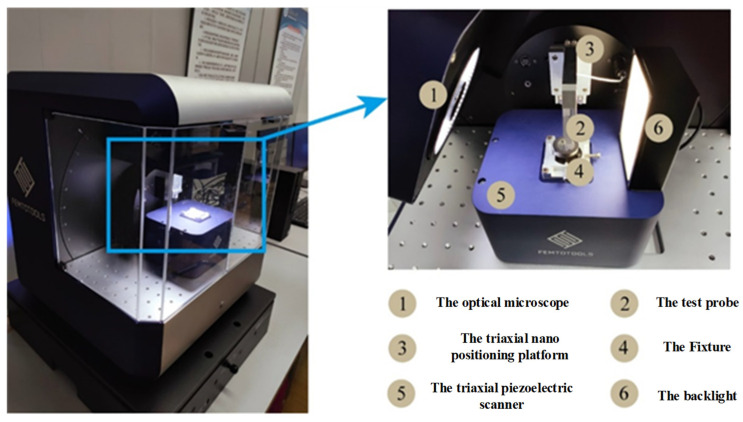
FT-MTA03 micromechanical testing and assembly system.

**Figure 11 micromachines-15-01020-f011:**
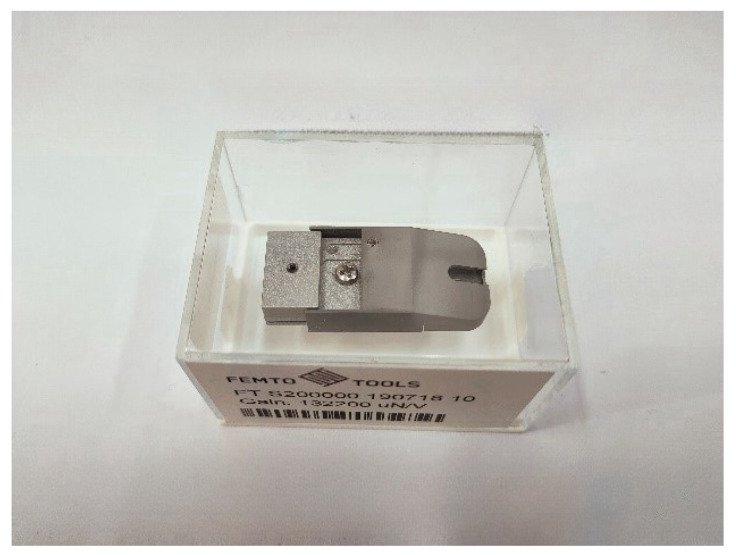
FT-S200000 micro-force sensor.

**Figure 12 micromachines-15-01020-f012:**
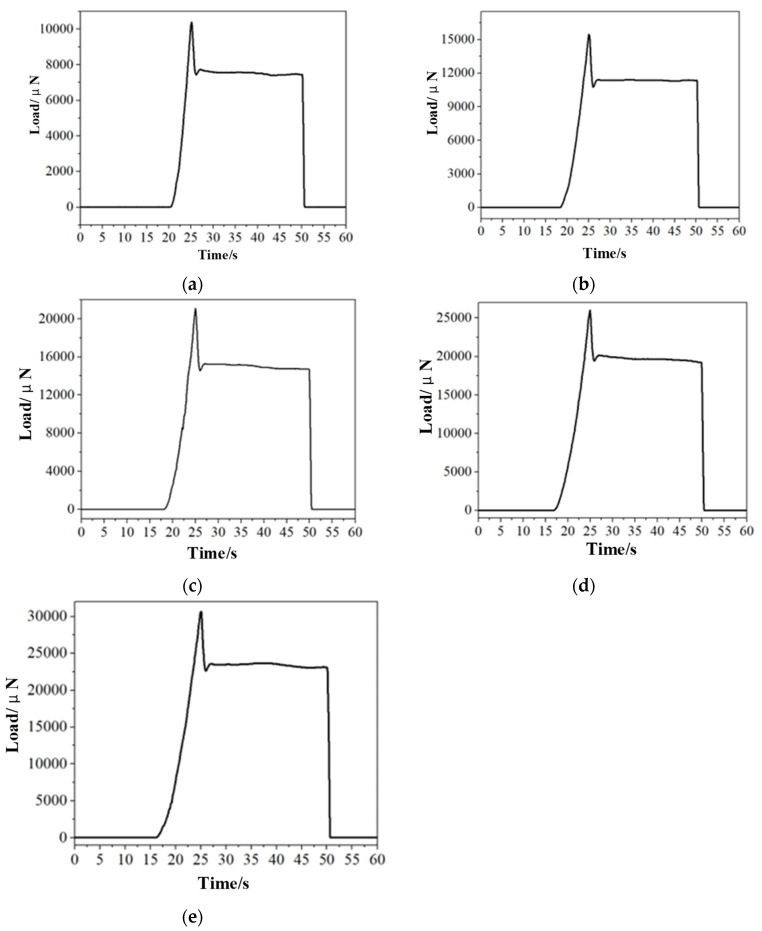
The load-time curve. (**a**) Load is 10 mN, (**b**) load is 15 mN, (**c**) load is 20 mN, (**d**) load is 25 mN, and (**e**) load is 30 mN.

**Figure 13 micromachines-15-01020-f013:**
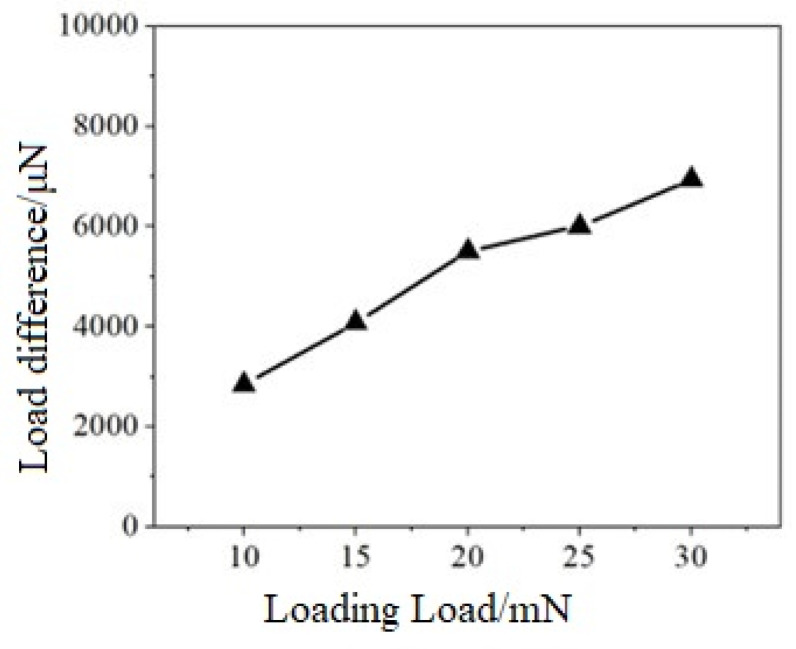
The load difference—load relationship diagram.

**Table 3 micromachines-15-01020-t003:** Analysis of the contents of constituent elements in the γ-TiAl alloy.

Element	(keV)	Sigma	Mass Percentage	Atomic Percentage
Al	1.486	0.15	37.76	51.86
Ti	4.508	0.30	62.24	48.14
Total amount			100	100

**Table 4 micromachines-15-01020-t004:** Maximum load value and displacement holding stage load value.

Loading Load (mN)	Maximum Load Value (mN)	Maintain Phase Load Value (mN)	Difference Value (mN)
10	10,391.6316	7552.1991	2839.4325
15	15,459.3222	11,371.9227	4087.3995
20	20,630.6382	15,128.5456	5502.0926
25	25,984.7146	19,979.7025	6005.0121
30	30,780.6481	23,847.7850	6932.8631

## Data Availability

Data are contained within this article.

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
