# Peer review of "Simulation and Experimental Study on Stress Relaxation Response of Polycrystalline γ-TiAl Alloy under Nanoindentation Based on Molecular Dynamics"

_micromachines, 2024, doi:10.3390/mi15081020_

Round 1
Reviewer 1 Report
Comments and Suggestions for Authors
Both nano-indentation and molecular dynamics simulations were carried out to examine the stress relaxation response of polycrystalline gamma-TiAl. A stress reduction is observed in the begining of the holding stage, and a continued though slow reducing during the holding process. With a higher preset indentation force, a larger reduction is observed. The findings is of help in understanding the stress relaxation and mechanical behavior of TiAl alloys. While the following points are suggested to the authors for consideration:
1) For the MD model, the initial model contains 6 grains. However, labels of G1-G5 is adopted, it seems that one grain is missing. Besides, during the indentation process, there is free surface along the z direction. With the creation of the free surface, will there still be 6 grains in the model?
2) Fig 1 illustrates the model created, while it seems that the axes shown in Fig 1(a) and (b) are mismatch with each other. Perhaps Fig 1(b) is a cross-section along x direction within the model?
3) EAM potential and Morse potentials are adopted to describe the interatomic interaction, while a proper citation of the original paper for the parameterization of the potentials are missing. Besides, Table 3 is a repeation of Table 1.
4) There are many misleading/confusing concepts used in the manuscript. For example, "atomic number" is a nomenclature to define the number of protons in the nucleus of an atom; while in the context the authors might intend to mean "the number of atoms". "balance" might be "equilibrate", "hot bath" should be "heat bath", "downward" and "upward" should be "loading" and "unloading", respectively.
5) The meaning of different colors in Fig 3 to Fig 5 is unclear to the audience.
6) The authors mentioned several times that "the atomic activity capacity" at the grain boundary is high. What is the physical meaning of "atomic activity capacity"? And, what is the "twin stress relaxation stage" referring to?
7) It is claimed that the released energy in G2 is transferred to G4 through the grain boundary. Is there any proof of this?
8) The two phase diagrams shown in Fig 7 is rudundant, Fig 8 and 10 are not quite necessary either. The subfigures in Fig 11 can also be merged into one figure.
Comments on the Quality of English LanguageThe usage of English is in general acceptable, while moderate editting is recommended.
Reviewer 2 Report
Comments and Suggestions for Authors
The article is devoted to simulation of stress relaxation response of TiAl alloy with nanoindentation using molecular dynamics as well as some simple experimental tests with Microforce Sensing Probes. In my view, the article was written in a perfunctory manner, it looks like to be just a mechanical compilation of different parts. I do not recommend the article for publishing. My comments are the following.
1. In page 2 the sentence “As a supplementary means of the experiment, etc” as well as the next one were repeated twice.
2. In page 3 there is no satisfactory explanation of Figure 1. What is the difference between grains G1-G5, how they were selected, what is their size, etc? Where is grain G6?
3. The same page it is written “the workspace is subjected to a pressing force of 50 m/s”. The force is measured in Newtons!
4. In Page 4 the authors gave Morse function parameters in Table 1. There is no clear description of this function, the column titles are “.” – full stop!, “Table .” .The introduction of the Morse potential function was done in Formula 4 on page 4. At the same page, the parameters of the function were indicated in Table 3. It is the same Table as Table 1 but with proper column titles. The authors have just mechanically connected two different text blocks without taking care how the full text looks like.
5. The picture in Figure 3 has to be explained. Is it a result of some simulation or just the consequences of the model described in Table 2?
6. The description of Figures 4 and 5 is not satisfactory. Again, what is the difference between grains? What is the meaning of abbreviations 4, A-A 321 in Figure 5a?
7. In Figure 7a the title of horizontal axis is not seen. There is no discussion of phase diagram in Figure 7. What was the reason to show it? The authors have just consider the sample with atomic ratio Ti:Al 1:1.
8. What is Sigma in Table 4?
9. What is the reason to show experimental sample in Figure 9 and conventional FT-MTA03 system in Figure 10? There is no useful information for a reader in these pictures.
10. The experimental parameters of indentation test do not correspond to simulation. Loading time was 25 seconds (50 ps in simulation), load values 10-30 mN, nNs in simulation, ets. There is no reason to compare experimental test and simulations.
11. The results in Figure 11 are conventional. I do not see there any useful information.
12. In Figure 12 there indicated Load difference at both axis. In Table 5 there indicated Loading load in column 1. What does it mean?
13. The authors have written that they compare experiment and simulation. Where is this comparison, what kind of information was compared? There is nothing about it in the article.
Comments on the Quality of English LanguageThere are numerous mistypings. In Figure abbreviations like (b), (d), etc must not be the full stop.
Round 2
Reviewer 1 Report
Comments and Suggestions for Authors
The authors have revised the manuscript partially according to the comments and suggestions of the reviewer, and the manuscript has been improved. It is however not quite satisfactory yet. The following points are suggested to the authors for consideration:
1) In the affiliation part, the first letter of the city name should be in upper case;
2) The change in number of the grains upon creation of surface is not known yet, and will this affect the analysis of the deformation process?
3) The 1st paragraph in section 2.2.1 and Equations 1-2 are not necessary. It is only valid for pair potential, it does not work for potentials like EAM shown in Eq. 3.
4) The necessity to show two phase diagrams in Fig. 7 is not well clarified.
Comments on the Quality of English LanguageThere is no substantial improvement on the usage of English in the revised version.
Reviewer 2 Report
Comments and Suggestions for Authors
The authors have improved the text but still there are weak points.
Page 4, Table 3 – the columns’ titles have to be corrected as it was indicated in the previous review.
Page 6 – reference to Table 3 is incorrect, it should be Table 2.
Page 7 – Figure 4 and corresponding text at the next page – the is no introduction and explanation what are abbreviations “a”, “b”, “c” neither in the Figure captions nor in the text.
Figure 4 (c,d), (e,f) – pictures at opposite sides are shifted relative to each other.
Page 9 – there appear “A-A inteface’, etc without any explanation in the text and Figure 5 captions.
Page 12 Figure 7 and corresponding text – The figure captions are wrong, one can see that in Figure 7a are the data for atomic Ti-Al ratio and in Figure 7b that for mole fraction. In both Figures are depicted phase diagrams starting from 600 oC. The experiment was done at room temperature. Since the authors do not discuss the properties of their sample Ti1Al1 at elevated temperatures there is no useful information in these pictures.
Page 12-13 before Table 3. The author’s statement “Sigma refers to the distance between two atoms” is obviously wrong because EDS cannot provide any structural information like interatomic distance.
Page 16 Figure 13 there must not be “load difference” notation on both axes. I guess there should be load value and load dereference.
There are also questions concerning scientific output of the article.
1) There is no explanation how Grains 1-6 were introduced in the model. The authors have just indicated that the grains have different locations. If there is any difference between the Grains in crystallographic orientations, how many atoms they have, how the intergrain boundaries were introduced? In a polycrystalline TiAl alloy there should be grains of different crystallographic orientations with wide-angle boundaries between them. Since in the article one uses periodic boundary conditions in x-y plain, there should be such grains in this plain. The authors should explain their model in more detail.
2) In Page 16 the author wrote “comparison between simulations results and test data indicates …”. There is no comparison between the simulation and experiment in the article. The simulation output is a) Load-time curve in Fig. 3, which is the direct consequence of the model, it was estimated for a model of 1.5 nm indentation depth; b) distribution pictures in Figs. 5-7. In the experiment one measures Load-time dependence for different load values (10-30 nN). The authors should explain in more detail what kind of data they compared, what is the result of their comparison, etc.
Comments on the Quality of English LanguageThere are some mistypings and errors that can be corrected.
Round 3
Reviewer 2 Report
Comments and Suggestions for Authors
The authors should correct the title of the second column in Table 1. I guess it should be "Ti-Al". I recommend the article for publishing.